# Information-Theoretic GAN Compression with Variational Energy-based Model

**Minsoo Kang**[1]   **Hyewon Yoo**[2]   **Eunhee Kang**[3]   **Sehwan Ki**[3]   **Hyong-Euk Lee**[3]   **Bohyung Han**[1,2]

[1]ECE & [2]IPAI, Seoul National University
[3]Samsung Advanced Institute of Technology (SAIT)
{kminsoo, yoohyewony, bhhan}@snu.ac.kr
{eunhee.kang, sh1004.ki, hyongeuk.lee}@samsung.com

## Abstract

We propose an information-theoretic knowledge distillation approach for the compression of generative adversarial networks, which aims to maximize the mutual information between teacher and student networks via a variational optimization based on an energy-based model. Because the direct computation of the mutual information in continuous domains is intractable, our approach alternatively optimizes the student network by maximizing the variational lower bound of the mutual information. To achieve a tight lower bound, we introduce an energy-based model relying on a deep neural network to represent a flexible variational distribution that deals with high-dimensional images and consider spatial dependencies between pixels, effectively. Since the proposed method is a generic optimization algorithm, it can be conveniently incorporated into arbitrary generative adversarial networks and even dense prediction networks, *e.g.*, image enhancement models. We demonstrate that the proposed algorithm achieves outstanding performance in model compression of generative adversarial networks consistently when combined with several existing models.

## 1 Introduction

Generative adversarial networks (GANs) [1] have accomplished impressive achievements on various computer vision tasks including image synthesis [2–5], image-to-image translation [6–9], video prediction [10–12], and many others. However, despite the outstanding performance, their applicability to resource-hungry systems, *e.g.*, edge or mobile devices, is limited due to high computational costs. To deal with this issue, GAN compression, a task reducing the size of generator networks, has recently drawn the attention of many machine learning researchers.

The straightforward way to compress generators is to directly employ traditional network compression techniques such as low-rank approximation [13–15], network quantization [16–19], pruning [20–25], and knowledge distillation [26–30]. However, the compression of generative models typically involves a higher-dimensional output space and requires matching the output distributions between the original and compressed models instead of the output instances given by the two models. Hence, as reported in [31–33], the synthesized images obtained from the compressed generators via the channel pruning [34, 23, 24] are often suboptimal—producing images with artifacts and suffering from image quality degradation, which implies that a naïve application of the methods designed for discriminative models with low dimensional output spaces is not effective for GAN compressions.

Existing approaches for compressing GANs [31, 35, 32, 36–38, 33, 39] often utilize knowledge distillation and manage to achieve competitive accuracy of student models. Specifically, these methods transfer the knowledge of a teacher network to a student model via intermediate representations,

36th Conference on Neural Information Processing Systems (NeurIPS 2022).

final outputs, and embedded features [40, 41]. To this end, the existing algorithms employ $\ell_1$ or $\ell_2$ norms to compute the difference between the representations of the two networks, but they are not necessarily effective for minimizing the distance between the two distributions.

We propose to compress GAN models by maximizing the mutual information for knowledge distillation between the outputs of a teacher and a student. For the compression of generative models, the use of the mutual information between two random variables makes more sense than the direct pixel-wise comparison between two outputs, which is the standard way to transfer knowledge from a teacher to a student in discriminative models. However, because the computation of the mutual information is generally intractable in the continuous domain, we employ a variational approach [42], where the lower bound of the mutual information is alternatively optimized using a variational distribution, an approximation for the conditional distribution of the teacher output given the student output. We further adopt an energy-based model for the flexible representation of the variational distribution and tighten the lower bound for effective knowledge transfer. Note that the proposed energy-based model is a convenient way to deal with high-dimensional data $e.g.$, images, and an effective tool to handle spatial dependency between pixels within an image. Combined with the existing approaches, our framework achieves substantial performance improvement on GAN compression tasks. The main contributions of our work are summarized below:

- We propose a novel energy-based framework of GAN compression for knowledge distillation, which maximizes the mutual information between a teacher and a student.
- The proposed energy-based approach is effective for representing a flexible variational distribution and can be incorporated into many GAN compression methods thanks to its generality.
- Experimental results verify that the proposed method improves performance consistently when combined with the existing methods.

The rest of the paper is organized as follows. Section 2 discusses the related work to GAN compression. The details of our approach are described in Section 3 and 4, and the experimental results are presented in Section 5. Finally, we conclude this paper in Section 6.

## 2 Related Work

### 2.1 Generative Adversarial Networks (GANs)

Generative Adversarial Networks (GANs) [1] consist of two components, a generator and a discriminator, and train the models using an adversarial loss; the generator aims to fool the discriminator by synthesizing realistic images from random noise samples while the discriminator attempts to distinguish the real data from the fake ones. Unlike the vanilla GANs, conditional GANs [43, 6, 7] control the image generation conditioned on additional inputs such as a class label [43] or an image to be translated into another image [6, 7] ($e.g.$, Horse $\rightarrow$ Zebra). The conditional GANs are not confined to the conditional image generation tasks, but applied to various areas such as image super-resolution [44], text-to-image synthesis [45], and image inpainting [46]. Since the computational costs of the conditional GANs are typically high in terms of FLOPs, the proposed GAN compression algorithm is mainly tested on those networks but we also verify the effectiveness of our approach on the unconditional case.

### 2.2 Knowledge Distillation

Knowledge distillation techniques aim to transfer knowledge from a powerful network, $i.e.$, teacher, to a compact network, $i.e.$, student, by matching logits [47], output distributions [26], intermediate activations [48], or attention maps [27]. Specifically, Hinton $et\ al.$ [26] learn the student representations by minimizing the relative entropy from its normalized softmax outputs to the teacher's while [47, 48] minimize the mean squared error between the representations of the teacher and student networks. Both Variational Information Distillation (VID) [49] and Contrastive Representation Distillation (CRD) [50] employ mutual information between teacher and student to estimate the similarity of the representations from the two networks. The former relies on a variational approach [42] for the tractable computation of mutual information between the two networks. On the other hand, the latter derives an InfoNCE-like loss function from the lower bound of the mutual information to learn the

representations of a student. However, VID maximizes the variational lower bound using a strong assumption about the variational distribution, a fully factorized Gaussian distribution, and CRD is difficult to be employed in generative models due to its contrastive learning objective, which is prone to be trivial in image generation tasks without semantic attribute annotations, *e.g.*, class labels. Our algorithm is related to VID and CRD since they all attempt to maximize the mutual information between the teacher and student networks, but has clear advantage over them in terms of the flexibility of the variational distribution and the applicability to generative models.

## 2.3 GAN Compression

GAN compression algorithms typically aim to reduce the size of the generator without considering the discriminator since it is not used for inference. Co-Evolution [31] utilizes an evolutionary algorithm [51] to reduce the channel size in the generator of CycleGAN [7] based on a fitness formulated based on FLOPs and the mean squared reconstruction error between the original uncompressed and pruned generators. GAN-Compression [35] employs a once-for-all network training scheme [52] combined with the reconstruction loss as well as the intermediate feature distillation loss [48], and selects the best performing subnetwork given a target budget. Compression and Teaching (CAT) [38] designs an efficient building block for generator, and prunes the network using a magnitude-based channel pruning method [24] while maximizing the similarity between the two networks based on centered kernel alignment [53]. Content-Aware GAN Compression (CAGC) [33] uses the feature distillation and reconstruction loss based only on the object parts of a generated image while additionally considering LPIPS [41] as a perceptual distance metric for synthesized images. On the contrary, Online Multi-Granularity Distillation (OMGD) [39] iteratively trains teacher networks with original conditional GAN objectives while optimizing a student network with the total variation loss [54] for spatial smoothness and the distillation loss for making the student mimic the final outputs and the intermediate feature maps of the teachers. Generator-discriminator Cooperative Compression (GCC) [55] also adopts the online distillation scheme while selecting channels in the original discriminator to match the reduced capacity of the student generator at each iteration. To validate the effectiveness of our framework, we present how the proposed algorithm can improve performance via the combinations with existing methods such as GCC, OMGD, CAGC, CAT, and GAN-Compression.

## 3 Variational GAN Compression with Energy-based Model

Our goal is to compress generators in GANs through knowledge distillation. Let $S$ and $T$ be random variables of the outputs from a student, $G^s(\cdot; \phi^s) : \mathcal{X} \to \mathcal{Y}$ and a teacher, $G^t(\cdot; \phi^t) : \mathcal{X} \to \mathcal{Y}$, respectively, where $\phi^s$ and $\phi^t$ are their model parameters. For knowledge transfer from the teacher to the student, we maximize the mutual information between $T$ and $S$, $I(T; S)$, which is given by

$$I(T; S) \equiv D_{\text{KL}}\Big(p\big(\boldsymbol{t}, \boldsymbol{s}\big) \,\|\, p\big(\boldsymbol{t}\big) \cdot p\big(\boldsymbol{s}\big)\Big) = H(T) + \mathbb{E}_{\boldsymbol{t}, \boldsymbol{s}}[\log p(\boldsymbol{t}|\boldsymbol{s})], \tag{1}$$

where $D_{\text{KL}}(\cdot\|\cdot)$ denotes the Kullback-Leibler (KL) divergence and $H(\cdot)$ means the entropy function. Note that all the expectations are taken over the joint data distribution, $p(\boldsymbol{t}, \boldsymbol{s})$, where $\boldsymbol{t}$ and $\boldsymbol{s}$ indicate realizations of the teacher and student output distributions, respectively.

A practical challenge in this objective is that the computation of the mutual information for multivariate continuous random variables is typically intractable. To tackle this issue, we adopt a variational method [42], where we derive the lower bound of the mutual information, $\tilde{I}(T; S)$, using a variational distribution, $q(\boldsymbol{t}|\boldsymbol{s})$, which is given by

$$\begin{aligned} I(T; S) &= H(T) + \mathbb{E}_{\boldsymbol{t}, \boldsymbol{s}}[\log q(\boldsymbol{t}|\boldsymbol{s})] + \mathbb{E}_{\boldsymbol{s}}[D_{\text{KL}}(p(\boldsymbol{t}|\boldsymbol{s})\|q(\boldsymbol{t}|\boldsymbol{s}))] \\ &\geq H(T) + \mathbb{E}_{\boldsymbol{t}, \boldsymbol{s}}[\log q(\boldsymbol{t}|\boldsymbol{s})] \\ &:= \tilde{I}(T; S). \end{aligned} \tag{2}$$

To maximize the mutual information, we increase the lower bound by minimizing the KL-divergence between the true and variational distributions, which is defined as

$$\triangle I(T; S) := I(T; S) - \tilde{I}(T; S) = \mathbb{E}_{\boldsymbol{s}}[D_{\text{KL}}(p(\boldsymbol{t}|\boldsymbol{s})\|q(\boldsymbol{t}|\boldsymbol{s}))]. \tag{3}$$

Variational Information Distillation (VID) [49] optimizes the variational lower bound by employing the fully factorized Gaussian distribution as the variational distribution $q(\boldsymbol{t}|\boldsymbol{s})$. However, as noted

in [56], the statistics of the natural images usually exhibit non-Gaussian properties. Furthermore, optimizing the lower bound using the fully factorized distribution encourages the student to generate blurry outputs [57, 46, 58]. To the contrary, we employ an energy-based model as a flexible distribution for the variational distribution, which is given by

$$q_\theta(\boldsymbol{t}|\boldsymbol{s}) = \frac{1}{Z_\theta(\boldsymbol{s})} \exp\Big(-E_\theta(\boldsymbol{t}, \boldsymbol{s})\Big),$$ (4)

where $E_\theta(\boldsymbol{t}, \boldsymbol{s}) : \mathcal{Y} \times \mathcal{Y} \to \mathbb{R}$ is the energy function defined by a deep neural network parametrized with $\theta$ and $Z_\theta(\boldsymbol{s})$ is the partition function. The variational distribution given by the energy-based model is effective for the student model to maximize the variational lower bound. In addition, differentiated from VID [49], our approach employs the variational distribution that considers the spatial dependency among pixels in $\boldsymbol{t}$ and $\boldsymbol{s}$.

Overall, we optimize the energy-based model, the student network, and the teacher network, which is achieved by the following iterative procedure:

1. **Fix $\{\phi^s, \phi^t\}$ and optimize $\theta$.**
   Under the fixed parameters $\{\phi^s, \phi^t\}$, minimize the objective in (3) with respect to $\theta$.
2. **Fix $\{\theta, \phi^t\}$ and optimize $\phi^s$.**
   Under the fixed parameters $\{\theta, \phi^t\}$, maximize the objective in (2) while minimizing the standard knowledge distillation losses with respect to $\phi^s$.
3. **Optimize $\phi^t$.**
   Minimize the standard conditional GAN loss [6, 7] with respect to $\phi^t$ using a training dataset. This step is only required for online knowledge distillation. Otherwise, the teacher model should be learned offline in advance.

These three steps are repeated until convergence.

## 4   Training Procedures

This section describes the optimization procedure of our approach for knowledge distillation, which iteratively maximizes the mutual information between a teacher and a student using the energy-based model.

### 4.1   Optimization of Energy-based Model

The first step of the iterative procedure is the optimization of the energy-based model for minimizing $\mathbb{E}_{\boldsymbol{s}}[D_{\mathrm{KL}}(p(\boldsymbol{t}|\boldsymbol{s})||q_\theta(\boldsymbol{t}|\boldsymbol{s}))]$ with respect to $\theta$ given the fixed model parameters $\{\phi^s, \phi^t\}$.

#### 4.1.1   Objective

Since the conditional data distribution $p(\boldsymbol{t}|\boldsymbol{s})$ is constant with respect to the model parameter $\theta$, the energy-based model minimizes the following objective:

$$\min_\theta \triangle I(T; S) \iff \min_\theta \mathbb{E}_{\boldsymbol{s}, \boldsymbol{t}}[-\log q_\theta(\boldsymbol{t}|\boldsymbol{s})].$$ (5)

To optimize the energy-based model using the standard stochastic gradient descent method, we derive the gradient of the objective in (5), which is given by

$$\nabla_\theta \mathbb{E}_{\boldsymbol{s}, \boldsymbol{t}}[-\log q_\theta(\boldsymbol{t}|\boldsymbol{s})] = \mathbb{E}_{\boldsymbol{s}, \boldsymbol{t}}\Big[\nabla_\theta E_\theta(\boldsymbol{t}, \boldsymbol{s})\Big] - \mathbb{E}_{\boldsymbol{s}}\mathbb{E}_{\tilde{\boldsymbol{t}} \sim q_\theta(\boldsymbol{t}|\boldsymbol{s})}\Big[\nabla_\theta E_\theta(\tilde{\boldsymbol{t}}, \boldsymbol{s})\Big],$$ (6)

where the second expectation of the right-hand side requires sampling from $q_\theta(\boldsymbol{t}|\boldsymbol{s})$ via a Markov Chain Monte Carlo (MCMC) method, for example, Gibbs sampling.

#### 4.1.2   Improved MCMC sampling

The main drawback of the MCMC methods is the high computational cost, so we rely on the Langevin dynamics, which leads to the following recursive sample generation:

$$\tilde{\boldsymbol{t}}^k = \tilde{\boldsymbol{t}}^{k-1} - \frac{\lambda}{2}\nabla_{\boldsymbol{t}} E_\theta(\tilde{\boldsymbol{t}}^{k-1}, \boldsymbol{s}) + w^k,$$ (7)

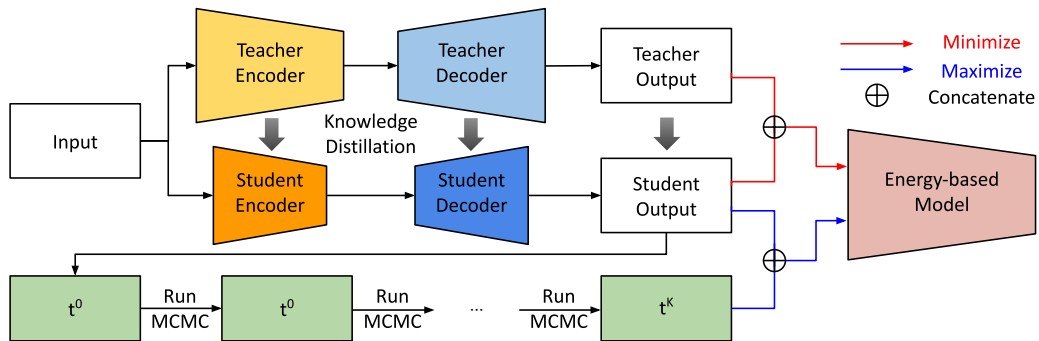

Figure 1: Overview of our knowledge distillation framework for GAN compression. It maximizes the variational lower bound of mutual information via an energy-based model.

where $w^k$ is a sample drawn from $\mathcal{N}(0, \lambda I)$ and $\lambda$ is a fixed step size. Theoretically, $\tilde{t}^k$ becomes identical to a sample drawn from $q_\theta(t|s)$ as $k \to \infty$ and $\lambda \to 0$ [59]. Following Short-Run MCMC [60], we also run $K$-step MCMC starting from an initial distribution, $q_0$, to approximate samples from $q_\theta(t|s)$, where $K$ is the number of MCMC steps.

### 4.1.3 Initial distribution

We initialize $\tilde{t}^0$ as the output of the student network for the acceleration of the MCMC optimization via the Langevin sampling process. This initialization is a good approximation to the model distribution; intuitively, since the student model is supposed to mimic the samples drawn from the data distribution, the optimization of $q_\theta(t|s)$ towards the data distribution $p(t|s)$ from the current student output would be more efficient. As a result, the proposed initialization strategy often results in improved performance of student models compared to the existing methods [61, 60, 62–65] even with a small number of MCMC steps. On the contrary, we can naturally adopt the data distributions of teacher networks, which are static in Contrastive Divergence (CD) [62] and dynamic in Persistent Contrastive Divergence (PCD) [63, 64], for setting $q_0$ while Short-Run MCMC [60] can initialize the variational distribution using the uniform noise distribution. Since the data distribution is prone to induce bias in the CD estimate [65] and the random uniform distribution is far from the data distribution, the existing methods require many steps of MCMC and consequently increase training time [60]. In addition, PCD updates all samples in the memory buffer, which leads to significantly increased memory overhead, especially in generation tasks.

### 4.1.4 Inference

In contrast to the previous approaches for energy-based models [60, 65, 64], we do not require any MCMC sampling for inference but generate images via a single forward process of the student generator. Therefore, the proposed algorithm incurs no extra computational cost at inference time.

## 4.2 Optimization of Student Generator

### 4.2.1 Loss for mutual information

The student generator learns to maximize the variational lower bound $\tilde{I}(T; S)$ in (2) with respect to $\phi_s$, where the objective function is formally given by

$$\max_{\phi^s} \tilde{I}(T; S) \iff \max_{\phi^s} H(T) + \mathbb{E}_{t,s}[\log q_\theta(t|s)] \iff \max_{\phi^s} \mathbb{E}_{t,s}[\log q_\theta(t|s)]. \tag{8}$$

Note that the entropy, $H(T)$, is independent of the parameters in the student network and the objective is equivalent to maximize the expected log likelihood as shown in the above equation. Since the joint distribution $p(t, s)$ depends on $\phi_s$, we employ a reparameterization trick using a function $G^s$. Then, the gradient of the objective function in (8) with respect to $\phi_s$ is derived as follows:

$$\nabla_{\phi^s} \mathbb{E}_{t,s}[-\log q_\theta(t|s)]$$
$$= \mathbb{E}_{\boldsymbol{x}}\left[\nabla_{\phi^s} E_\theta\big(G^t(\boldsymbol{x}; \phi^t), G^s(\boldsymbol{x}; \phi^s)\big)\right] - \mathbb{E}_{\boldsymbol{x}}\mathbb{E}_{\tilde{\boldsymbol{t}} \sim q_\theta\big(\boldsymbol{t}|G^s(\boldsymbol{x};\phi^s)\big)}\left[\nabla_{\phi^s} E_\theta\big(\tilde{\boldsymbol{t}}, G^s\big(\boldsymbol{x}; \phi^s\big)\big)\right]. \tag{9}$$

Table 1: Performance of VEM in comparison with the state-of-the-art compression methods for the Pix2Pix model using the U-Net baseline on the Edges → Shoes and Cityscapes datasets. Methods with an asterisk (*) denote our reproductions given by running the official codes.

| Dataset | Method | MACs | # of parameters | FID ($\downarrow$) | mIoU ($\uparrow$) |
|---------|--------|------|-----------------|---------|----------|
| Edges → Shoes | Original [6] | 18.60G (1.0$\times$) | 54.40M (1.0$\times$) | 34.31 | - |
| | DMAD [37] | 2.99G (6.2$\times$) | 2.13M (25.5$\times$) | 46.95 | - |
| | OMGD* [39] | 1.22G (15.3$\times$) | 3.40M (16.0$\times$) | 27.39 | - |
| | OMGD* [39] + VID [49] | 1.22G (15.3$\times$) | 3.40M (16.0$\times$) | 28.02 | - |
| | OMGD* [39] + VEM (Ours) | 1.22G (15.3$\times$) | 3.40M (16.0$\times$) | **24.59** | - |
| Cityscapes | Original [6] | 18.60G (1.0$\times$) | 54.40M (1.0$\times$) | - | 42.71 |
| | DMAD [37] | 3.96G (4.7$\times$) | 1.73M (31.4$\times$) | - | 40.53 |
| | GCC [55] | 3.09G (6.0$\times$) | - | - | 42.88 |
| | OMGD* [39] | 1.22G (15.3$\times$) | 3.40M (16.0$\times$) | - | 47.52 |
| | OMGD* [39] + VID [49] | 1.22G (15.3$\times$) | 3.40M (16.0$\times$) | - | 47.79 |
| | OMGD* [39] + VEM (Ours) | 1.22G (15.3$\times$) | 3.40M (16.0$\times$) | - | **49.39** |

Note that we do not need to perform MCMC additionally to obtain the samples from $q_\theta(\boldsymbol{t}|G^s_{\phi_s}(\boldsymbol{x}))$ because they are already available from the training procedure for the energy-based model. Figure 1 illustrates the overview of our framework for maximizing the mutual information.

#### 4.2.2 Algorithm-specific loss

The proposed variational lower bound maximization framework using the energy-based model is incorporated into model compression algorithms based on knowledge distillation such as OMGD [39], GCC [55], GAN-Compression [35], CAT [38], and CAGC [33]. The brief descriptions about these methods are presented in Section 2.3, and we discuss the details of the algorithm-specific losses in the supplementary document.

#### 4.2.3 Combined loss

The full objective function for the student network is given by

$$\mathcal{L}^s = \mathcal{L}_{\text{algo}} - \lambda_{\text{MI}}\tilde{I}(T; S), \tag{10}$$

where $\mathcal{L}_{\text{algo}}$ is the algorithm-specific loss and $\tilde{I}(T; S)$ is the variational lower bound defined in (8) while $\lambda_{\text{MI}}$ is a hyperparameter balancing the two terms.

### 4.3 Optimization of Teacher Generator

The teacher generator is trained by the standard method of the corresponding GAN algorithm, *e.g.*, Pix2Pix [6] and CycleGAN [7]. Optionally, the teacher model learns to minimize the $\ell_1$ distance between the generated image and the ground-truth when the ground-truth is available. In the case of the unsupervised image-to-image translation tasks, the cycle consistency loss [7] based on the $\ell_1$ distance may enforce the instance mapping between two domains. To sum up, the teacher network is optimized by the adversarial loss, denoted by $\mathcal{L}_{\text{GAN}}$, and the reconstruction loss, $\mathcal{L}_{\text{rec}}$, as follows:

$$\min_{G^t} \max_{D^t} \mathcal{L}_{\text{GAN}} + \lambda_{\text{rec}}\mathcal{L}_{\text{rec}}, \tag{11}$$

where $\lambda_{\text{rec}}$ is a hyperparameter to balance the two loss terms. Note that, for unconditional GANs, there is nothing to reconstruct and we train the network based only on the GAN loss.

## 5 Experiments

To validate the effectiveness of the proposed compression algorithm based on the variational energy-based model, referred to as VEM, we conduct extensive experiments on the standard datasets using various architectures. Since VEM is a model-agnostic algorithm component, we incorporate the module into existing GAN compression approaches and the combined models are expressed in the form of "[Algorithm Name] + VEM" throughout this section.

Table 2: Performance comparisons between VEM and the state-of-the-art compression methods in CycleGAN model with the ResNet backbones.

| Dataset | Method | MACs | # of parameters | FID ($\downarrow$) |
|---|---|---|---|---|
| Horse → Zebra | Original [7] | 56.80G (1.0×) | 11.30M (1.0×) | 61.53 |
| | Co-Evolution [31] | 13.40G (4.2×) | - | 96.15 |
| | GAN-Slimming [36] | 11.30G (23.6×) | - | 86.09 |
| | Auto-GAN-Distiller [32] | 6.39G (8.9×) | - | 83.60 |
| | GAN-Compression [35] | 2.67G (21.3×) | 0.34M (33.2×) | 64.95 |
| | DMAD [37] | 2.41G (23.6×) | 0.20M (40.0×) | 62.96 |
| | CAT [38] | 2.55G (22.3×) | - | 60.18 |
| | GCC [55] | 2.40G (23.7×) | - | 59.31 |
| | OMGD* [39] | 1.41G (40.3×) | 0.14M (82.5×) | 57.14 |
| | OMGD* [39] + VEM (Ours) | 1.41G (40.3×) | 0.14M (82.5×) | **50.83** |
| Summer → Winter | Original [7] | 56.80G (1.0×) | 11.30M (1.0×) | 79.12 |
| | Co-Evolution [31] | 11.10G (5.1×) | - | 78.58 |
| | Auto-GAN-Distiller [32] | 4.34G (13.1×) | - | 78.33 |
| | DMAD [37] | 3.18G (17.9×) | 0.30M (37.7×) | 78.24 |
| | OMGD* [39] | 1.41G (40.3×) | 0.14M (82.5×) | 75.20 |
| | OMGD* [39] + VEM (Ours) | 1.41G (40.3×) | 0.14M (82.5×) | **74.04** |

Table 3: Performance comparisons between VEM and the existing compression algorithms for unconditional GANs using SAGAN and StyleGAN2 on CelebA and FFHQ, respectively.

| Model | Dataset | Method | MACs | Compression rate | FID ($\downarrow$) |
|---|---|---|---|---|---|
| SAGAN [4] | CelebA | Original | 23.45M | - | 24.87 |
| | | Slimming [24] | 15.45M | 34.12% | 36.60 |
| | | GCC* [55] | 15.45M | 34.12% | 27.91 |
| | | GCC* [55] + VEM (Ours) | 15.45M | 34.12% | **25.27** |
| StyleGAN2 [66] | FFHQ | Original | 45.1B | - | 4.50 |
| | | GAN-Slimming [36] | 5.0B | 88.91% | 12.40 |
| | | CAGC [33] | 4.1B | 90.91% | 7.90 |
| | | CAGC [33] + VEM (Ours) | 4.1B | 90.91% | **7.48** |

## 5.1 Datasets and Models

**Image-to-image translation** We adopt Pix2Pix [6] on the Edges → Shoes [67] and Cityscapes [68] datasets while taking CycleGAN [7] on Horse → Zebra [7] and Summer → Winter [7]. Note that Pix2Pix is the image-to-image translation network for paired data while CycleGAN is mainly proposed for unpaired image-to-image translation. Following the original models [6, 7], we employ generators based on U-Net [69] and ResNet [70] for Pix2Pix and CycleGAN, respectively. For all the datasets, images are resized to $256 \times 256$ before feeding them into the models.

**Image generation** To evaluate the performance of our approach for unconditional GANs, we run experiments with StyleGAN2 [66] on Flickr-Faces-HQ (FFHQ) [71] and Self-Attention GAN (SAGAN) [4] on CelebA [72]. As a preprocessing, we resize input images to $64 \times 64$ for CelebA and $256 \times 256$ for FFHQ.

## 5.2 Evaluation Metrics

Following the protocol of the previous approaches, we report the Fréchet Inception Distance (FID) [73] scores on Edges → Shoes, Horse → Zebra, Summer→Winter, CelebA, and FFHQ while the mean Intersection over Union (mIoU) is selected for Cityscapes. The FID score is widely used for assessing the quality of the generated images by measuring the distance between the distributions of embedding features extracted from generated and real images, which are given by the Inception V3 model [74] pretrained on ImageNet. In the case of the Cityscapes dataset, we adopt DRN-D-105 [75] as a model to perform semantic segmentation on the translated images and derive mIoU scores for the evaluation of image-to-image translation networks.

### 5.3 Implementation Details

The proposed algorithm is implemented in PyTorch [76] based on the publicly available code[1]. Additionally, we incorporate the proposed mutual information maximization framework into existing approaches by using the official codes of GAN-Compression[2] [35], CAT[3] [38], CAGC[4] [33], and GCC[5] [55] to validate its benefit and generality. For fair comparisons, we do modify neither experimental settings nor network structures of the compared algorithms. Specifically, we set the batch size to 4 for Pix2Pix, 1 for CycleGAN, 64 for SAGAN, and 16 for StyleGAN2 while we adopt the Adam optimizer with an initial learning rate of 0.0002, which decays to zero linearly.

When we train our energy-based model, we employ the Adam optimizer with a learning rate of 0.0001. Note that we only run 10 steps of Langevin dynamics to draw samples from the energy-based model to save training time. We set the standard deviation of random noise to 0.005 and use a step size of 100 for each gradient step of Langevin dynamics for all datasets except for Cityscapes and FFHQ. For these two datasets, we use a step size of 50 instead. Details about the hyperparameter settings are provided in the supplementary document. In the case of paired datasets such as Edges $\rightarrow$ Shoes and Cityscapes, we maximize the mutual information between the outputs given by the student model and the real outputs instead of the teacher outputs.

### 5.4 Results

**Pix2Pix**    Table 1 presents the results from the Pix2Pix models on Edges $\rightarrow$ Shoes and Cityscapes, where the proposed algorithm, denoted by OMGD + VEM, is compared with existing approaches including the vanila OMGD [39]. As shown in the table, OMGD + VEM achieves the best performance in terms of FID and MAC among all compared algorithms on Edges $\rightarrow$ Shoes. On Cityscapes, OMGD + VEM presents the best mIoU by large margins even with the highest compression rate. Note that we report the results from OMGD based on our reproductions using the official code, which are denoted by asterisks (*) in all tables.

**CycleGAN**    We test our compression algorithm for CycleGAN on Horse $\rightarrow$ Zebra and Summer $\rightarrow$ Winter. As shown in Table 2, OMGD + VEM outperforms all the compared methods in both datasets, especially by large margins in Horse $\rightarrow$ Zebra.

**Unconditional GANs**    We compress SAGAN and StyleGAN2 using GCC [55] and CAGC [33], respectively. Table 3 illustrates that the proposed method is also effective to improve performance of the unconditional GAN compression techniques. In particular, our approach also works well with a large-scale generator such as StyleGAN2.

### 5.5 Ablation Study about Mutual Information Maximization Techniques

To maximize the mutual information between the teacher and student networks, one can use a fully-factorized Gaussian distribution for the variational distribution as in VID [49] or employ a contrastive loss similar with infoNCE [77] following the strategy in CRD [50]. To validate the effectiveness of our energy-based models, VEM, in comparison with VID and CRD, we perform an ablation study, where the three add-ons are tested through their integrations into the baselines such as OMGD [39], GAN-Compression [35], CAT [38], GCC [55], and CAGC [33].

Table 4 demonstrates that VEM consistently outperforms VID and CRD in all the tested GAN compression algorithms. This is because, compared to VID, we derive a tighter lower bound of the mutual information theoretically and consider spatial dependencies among pixels thanks to the more flexible variational distribution based on the energy function. On the other hand, the application of CRD to generative models may not be beneficial since instance-level contrastive losses are less effective due to large gaps between positive and negative pairs.

---

[1]https://github.com/bytedance/OMGD

[2]https://github.com/mit-han-lab/gan-compression

[3]https://github.com/snap-research/CAT

[4]https://github.com/lychenyoko/content-aware-gan-compression

[5]https://github.com/sjleo/gcc

Table 4: Ablation studies about the approaches to maximizing mutual information. We compare the methods relying on the proposed energy-based model (VEM) with the approaches based on a fully-factorized Gaussian distribution (VID) and an InfoNCE-like contrastive loss (CRD). This experiment is conducted on the Horse $\rightarrow$ Zebra, Summer $\rightarrow$ Winter, CelebA, and FFHQ datasets.

| Model | Dataset | Method | MACs | FID ($\downarrow$) |
|---|---|---|---|---|
| CycleGAN [7] | Horse $\rightarrow$ Zebra | OMGD* [39] | 1.41G | 57.14 |
| | | OMGD* [39] + VID [49] | 1.41G | 65.73 |
| | | OMGD* [39] + CRD [50] | 1.41G | 70.59 |
| | | OMGD* [39] + VEM (Ours) | 1.41G | **50.83** |
| | | CAT* [38] | 2.56G | 64.79 |
| | | CAT* [38] + VID [49] | 2.56G | 68.71 |
| | | CAT* [38] + CRD [50] | 2.56G | 67.69 |
| | | CAT* [38] + VEM (Ours) | 2.56G | **52.44** |
| | | GAN-Compression* [35] | 2.55G | 59.13 |
| | | GAN-Compression* [35] + VID [49] | 2.55G | 62.36 |
| | | GAN-Compression* [35] + CRD [50] | 2.55G | 60.27 |
| | | GAN-Compression* [35] + VEM (Ours) | 2.55G | **50.01** |
| | Summer $\rightarrow$ Winter | OMGD* [39] | 1.41G | 75.20 |
| | | OMGD* [39] + VID [49] | 1.41G | 74.39 |
| | | OMGD* [39] + CRD [50] | 1.41G | 74.54 |
| | | OMGD* [39] + VEM (Ours) | 1.41G | **74.04** |
| SAGAN [4] | CelebA | GCC* [55] | 15.45M | 27.91 |
| | | GCC* [55] + VID [49] | 15.45M | 26.57 |
| | | GCC* [55] + CRD [50] | 15.45M | 30.93 |
| | | GCC* [55] + VEM (Ours) | 15.45M | **25.27** |
| StyleGAN2 [66] | FFHQ | CAGC [33] | 4.1B | 7.90 |
| | | CAGC [33] + VID [49] | 4.1B | 7.51 |
| | | CAGC [33] + CRD [50] | 4.1B | 7.65 |
| | | CAGC [33] + VEM (Ours) | 4.1B | **7.48** |

Table 5: Sensitivity analysis for $\lambda_{MI}$ on the Cityscapes dataset using Pix2Pix when VEM is combined with OMGD.

| Dataset | Method | $\lambda_{MI}$ | MACs | # of parameters | mIoU ($\uparrow$) |
|---|---|---|---|---|---|
| Cityscapes | OMGD* [39] | – | 1.22G (15.3×) | 3.40M (16.0×) | 47.52 |
| | OMGD* [39] + VEM (Ours) | 0.05 | 1.22G (15.3×) | 3.40M (16.0×) | 47.56 |
| | OMGD* [39] + VEM (Ours) | 0.10 | 1.22G (15.3×) | 3.40M (16.0×) | 48.90 |
| | OMGD* [39] + VEM (Ours) | 0.20 | 1.22G (15.3×) | 3.40M (16.0×) | **49.39** |

## 5.6 Effect of $\lambda_{MI}$ for Mutual Information

We analyze the effect of the balancing parameter $\lambda_{MI}$ using the Pix2Pix algorithm, which is tested on the Cityscapes dataset, when VEM is combined with OMGD [39]. As shown in Table 5, VEM is not sensitive to the hyperparameter and improves the performance of OMGD.

## 5.7 Qualitative Results

Figure 2 visualizes generated images given by the original uncompressed backbones of Pix2Pix and CycleGAN, where we also present the images synthesized by the compressed counterparts using OMGD and OMGD + VEM. Pix2Pix is tested on the Edges $\rightarrow$ Shoes and Cityscapes datasets, and CycleGAN is evaluated on the Horse $\rightarrow$ Zebra and Summer $\rightarrow$ Winter datasets. The results imply that our approach is effective to enhance generation quality. For example, OMGD often produces unusual artifacts in the background with the Cityscapes dataset, which are significantly reduced in OMGD + VEM. Figure 3 depicts generation results given by unconditional GANs on CelebA and FFHQ. Compared with the uncompressed versions, the proposed method preserves or often improves image quality even with high compression rates. Also, our approach synthesizes images with better visual quality when combined with the previous GAN compression methods such as GCC and CAGC.

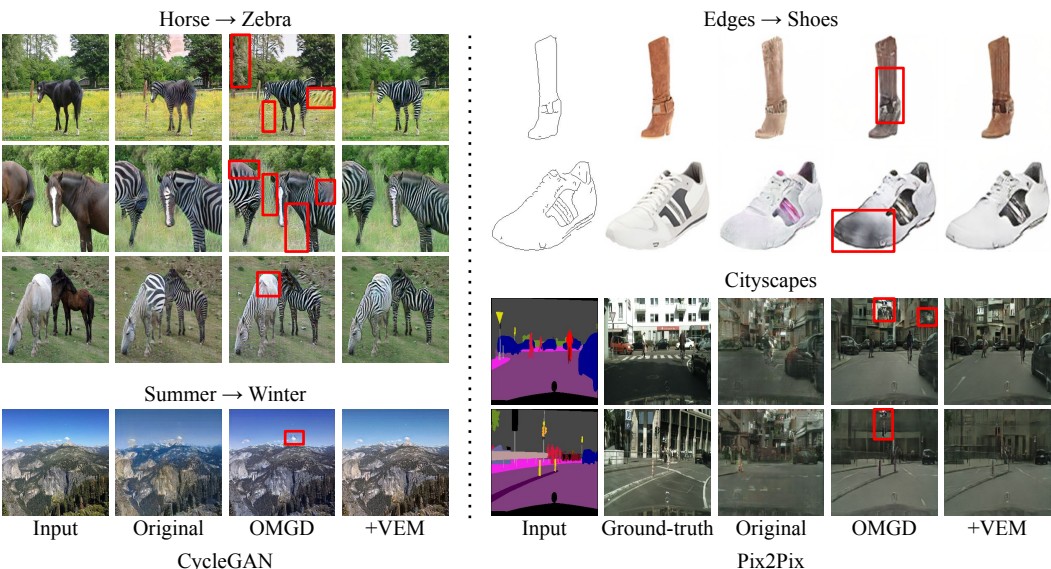

Figure 2: Qualitative results of CycleGAN on Horse → Zebra and Summer → Winter (left) and Pix2Pix on Edges → Shoes and Cityscapes (right). Note that "Original" represents the images generated by the uncompressed backbones.

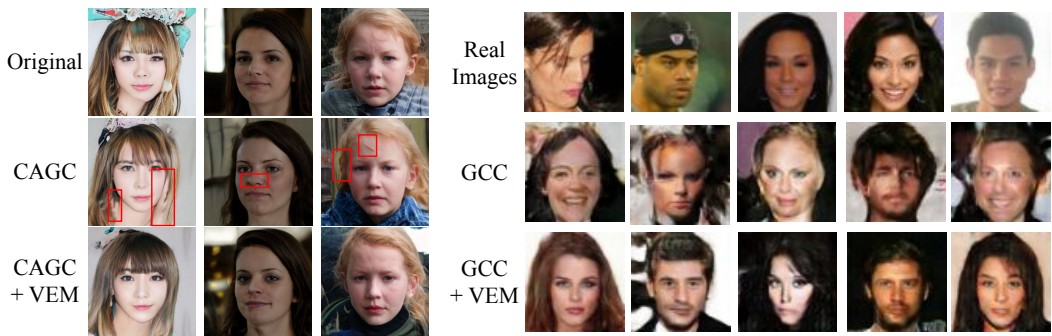

Figure 3: Qualitative results of StyleGAN2 using on the FFHQ dataset (left) and SAGAN on the CelebA dataset (right). Note that "Real Images" denotes data samples in CelebA while "Original" represents the images generated by the uncompressed StyleGAN2.

## 6   Conclusion

We presented a novel GAN compression method via knowledge distillation based on mutual information, which maximizes the lower bound of the mutual information by incorporating an energy-based variational distribution. Our energy-based model is straightforward to be incorporated into various existing model compression algorithms for generative models. It provides a more flexible variational distribution, which leads to theoretically tighter lower bounds and facilitates the consideration of the dependency between pixels. The proposed algorithm demonstrates outstanding performance in various GAN compression scenarios on multiple datasets compared to the state-of-the-art approaches.

**Acknowledgments**   This work was partly supported by SAIT, Samsung Electronics Co., Ltd, the Bio & Medical Technology Development Program of the National Research Foundation (NRF) funded by the Korea government (MSIT) [No. 2021M3A9E4080782], and Institute of Information & communications Technology Planning & Evaluation (IITP) grant funded by the Korea government (MSIT) [No. 2021-0-01343, No. 2022-0-00959].

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
