# OpenReview forum: "Information-Theoretic GAN Compression with Variational Energy-based Model"
_NeurIPS.cc/2022/Conference — NeurIPS 2022 Accept_

### Official Review · Reviewer_tn1T · 2022-07-08

**Rating:** 4
**Confidence:** 5
**Soundness:** 2 fair
**Presentation:** 3 good
**Contribution:** 3 good

**Summary:**

This paper proposes a knowledge distillation approach to accelerate GANs via maximizing mutual information between the teacher model and the student model. While mutual information (MI) between two continuous distribution is intractable, the paper alternatively maximize the lower bound of MI. The method is evaluated on both conditional and unconditional GANs with a noticeable numeric improvement.

**Questions:**

See weakness.

**Limitations:**

Yes.

**Strengths And Weaknesses:**

Strengths:
1. While VID has been proposed for classification model distillation, maximizing mutual information between teacher and student for GANs is more challenging and bears certain novelty.
2. The motivation is clear and the technical details look correct.
3. Experiments are carried out on multiple tasks and show good quantitative results.

Weakness:
1. My primary concern lies in the qualitative results. In fact, there is still not a good metric that really quantifies GAN's performance and thus the qualitative results are crucial to reflect the effectiveness of the approach. Looking at Fig. 2 of the paper, it's hard to tell whether OMGD or OMGD + VEM bears better quality. In addition, it seems that +VEM does not push the outputs from student to look similar to the teacher's. Why is that the case, as the mutual information is maximized during the distillation? It would be good to include figures or quantitative metrics (between teacher and student outputs, not FID) demonstrating the effectiveness of the distillation.
2. The approach is only evaluated on GANs generating 256px images. However, prior method like CAGC has been applied for 1024px StyleGAN2. How would the method perform on high-resolution synthesis?
3. The method uses an additional energy-based neural network and requires back-propagation to optimize its parameters, which is non-trival overhead. A computational time profiling on the additional module should be presented.

---

> ### Author Response · Authors · 2022-08-02
> **Author Response**
>
> We truly thank you for your constructive and positive comments and below are our responses to the main questions.
>
> Q1. Discussion about the visual quality presented in figure 2
>
> A. We partially agree with you, however our method provides much better quality on the horse $\rightarrow$ zebra dataset, where our method significantly outperforms OMGD. For other datasets in figure 2, since the performance improvement is relatively smaller, it may be hard to mention that our method bears better visual quality. Also, we clarify that ‘Original’ indicates not a teacher network but a network provided by the official checkpoints given by [A1, A2]. Also, requested by Reviewer tn1T, we measured the perceptual distance using feature reconstruction loss [A3] between the two output distributions of the teacher and student networks for quantitative metrics to demonstrate the effectiveness of VEM. In case of the paired datasets using Pix2Pix, we did not perform the experiments to measure the distance since VEM maximizes the mutual information between the student outputs and ground-truth ones as mentioned in the main paper. As a result, it is empirically validated that VEM does push the outputs given by the student to look similar to the one given by the teacher as presented in Table A3.
>
> Table A3: Averaged perceptual distance [A3] over the validation dataset between the two output distributions of the teacher and student networks using CycleGAN.
>
> | Dataset | Method  | Perceptual distance [A3]  |
> |:----------------:|:---------------:|:---------------:|
> |    Horse $\rightarrow$ Zebra   |  OMGD*   | 0.3805 |
> |   Horse $\rightarrow$ Zebra    |   OMGD* + VEM   | 0.3721 |
> |      Summer $\rightarrow$ Winter     | OMGD* | 0.1843 |
> |      Summer $\rightarrow$ Winter      |  OMGD* + VEM  | 0.1783 |
>
>
> Q2. Additional experiments for 1024x1024 resolutions using StyleGAN2
>
> A. We appreciate you for a good suggestion, but it is really hard to perform the experiment you suggested since it takes an even longer time to compress a Stylegan2 for 1024 pixel image generation tasks. We believe that VEM can work well for generative models for 1024 pixels since our method is theoretically validated and not limited to the ones for 256 pixels. We will perform the experiment and will add the results in the final version if our paper is accepted.
>
> Q3. Computational overhead about VEM
>
> A. Please refer to our response to Q3 of Reviewer 6hcq.
>
>
> Reference
>
> [A1] P. Isola et al., Image-to-Image Translation with Conditional Adversarial Networks, CVPR 2017.
>
> [A2] J.Y. Zhu et al., Unpaired Image-to-Image Translation using Cycle-Consistent Adversarial Network, ICCV2017.
>
> [A3] J. Johnson et al., Perceptual Losses for Real-Time Style Transfer and Super-Resolution, ECCV 2016.

---

> > ### Author Response · Authors · 2022-08-08
> > **After Rebuttal**
> >
> > Dear Reviewer  tn1T,
> >
> > Because the end of discussion period is approaching, we kindly ask you whether our response is helpful to clarify you or not.
> >
> > Also, if you have any questions or additional comments, please do not hesitate to contact us.
> >
> > Once again, thank you for your time and efforts to review our paper.
> >
> > Best wishes,
> >
> > Authors

---

> > > ### Comment · Reviewer_tn1T · 2022-08-08
> > > **Reviewer Response**
> > >
> > > Thanks for your response. The added table looks good to me. However, I still think the qualitative results is an issue of the work. Different from discriminative models, quantitative metrics weigh much less to represent practicality of GANs. The paper will get stronger if you can show VEM does push the student to mimic teacher better in image translation, image embedding, and image generation. From Fig.2 and Fig.3, it's still hard to tell how +VEM improves upon OMGD and CAGC.

---

> > > > ### Author Response · Authors · 2022-08-09
> > > > **After Rebuttal**
> > > >
> > > > Upon request by Reviewer tn1T, we present more qualitative results of VEM, OMGD, and CAGC in Figure 13, 14, 15 in the new supplementary document (Rebuttal_VEM.pdf). As seen in the presented images, VEM achieves qualitative better results than OMGD and CAGC and tends to generate the images that are visually similar to teacher images. Note that we did not run additional experiments to prepare the new supplementary file and chose image pairs (out of the existing results made for our initial submission) that have larger visual quality differences.

---

### Official Review · Reviewer_6hcq · 2022-07-13

**Rating:** 7
**Confidence:** 4
**Soundness:** 3 good
**Presentation:** 3 good
**Contribution:** 3 good

**Summary:**


Sampling generative networks on devices with limited computational capabilities imposes limits on complexity of these networks. The concept of distillation provides one approach to obtaining networks whose performance gracefully degrades with network complexity.

Paper proposes a new distillation technique for generative adversarial networks. The method uses a tractable lower bound on mutual information between student and teacher generative networks.

Authors introduce a family of methods which optimize variational lower bound on the mutual information between teacher and student network. The optimizer alternates between updating variational parameters, student network’s parameters and optionally teacher network’s parameters. This approach combines well with existing network compression algorithms.

**Questions:**

What is the interpretation of vertical arrows in Fig1?

Backbone algorithms? Reference?

What is the size of parameterization of the q(t|s) network for different tasks (architecture from supp inf Fig.4)?
What is the computational cost of tacking VEM on top of other distillation/compression methods?

Is VEM applicable in absence of a distillation method to piggy back on?

**Limitations:**

Due to presence of MCMC steps and need to track auxiliary Langevin dynamics variables there is an increased space and computational cost. The degree to which this cost can be decreased (or increased) and how that impacts the performance of the method is not quite clear.

**Strengths And Weaknesses:**

The proposed method can be combined with existing compression algorithms. It improves performance measured by FID and mIoU for fixed number of parameters in a network compared to compression algorithms it augments. The paper is clear with few small corrections. Code is readily available.


Clarity: specify that p(t,s) = p(t|x) p(s|x)p(x) since mutual information stems from reuse of code between student and teacher.
It might be helpful to provide an example of perfect q(t|s) that keeps the bound tight -- p(x|s)p(t|x) -- and consists of decoder of students output and teacher's encoder.

Clarity:``student generator learns to maximize the variational lower bound'' might be more helpful rewritten as ``generator is trained by maximizing …''  The point here is that: Optimization algorithm optimizes, learning algorithm learns (if bug free and fed the right data), while the student generator ... generates samples. Crucially student generator does not learn how to maximize.

Visuals in Figure 1 are a little difficult to understand. Arrows seem to have inconsistent meanings. Some arrows are simply meant as feeding forward outputs from one stage to another. Others are more abstract like the vertical arrows connecting teacher and student networks. Perhaps the intent was to refer to distillation methods embedded in the overall framework. However, the arrows themselves do not have a mathematical or algorithmic analog in the paper. This left me wondering if I missed some piece of the machinery.

FID, MACs, mIoU and any other abbreviations should be defined and referenced.

---

> ### Author Response · Authors · 2022-08-02
> **Author Response**
>
> We truly thank you for your constructive and positive comments and below are our responses to the main questions.
>
> Q1. Presentation issues related to clarification, figure 1, and definition of abbreviations
>
> A.  We will carefully revise the paper to reflect your comments in the final version. For $q(t|s)$, it is impossible to check whether the variational distribution is tight or not because the true distribution is unknown. In figure 1, as you mentioned, the vertical arrows indicate the knowledge distillation employed in the previous methods.
>
> Q2. What is the interpretation of vertical arrows and backbone algorithms?
>
> A. In figure 1, as you mentioned, the vertical arrows indicate the knowledge distillation employed in the previous methods. For the knowledge distillation, the details are presented in Section A.1 of the supplementary material and please refer to it.
>
> Q3. The size of energy-based model and computational cost of VEM
>
> A.  For the size of the energy-based model, it consumes 0.12G MACs for the Horse $\rightarrow$ Zebra dataset while the model requires 1.95G MACs for other datasets. Also, our method requires an extra 0.5x ∼1.5x training cost depending on datasets and models. Although the presence of MCMC in VEM incurs an extra training cost, the cost can be reduced by decreasing the number of MCMC steps. When the number of MCMC steps is reduced from 10 (used in all experiments) to 5 combined with OMGD, the FID score on the Horse $\rightarrow$ Zebra dataset is increased from 50.83 to 52.04, which still outperforms the baseline algorithm. Therefore, there is a trade-off between the performance and the training time and we will add the discussions in the final version.
>
> Q4. Is VEM applicable in absence of a distillation method?
>
> A. Yes, VEM is applicable when the previous methods are absent. But, we believe that VEM is more effective when combined with the methods

---

> > ### Comment · Reviewer_6hcq · 2022-08-09
> > **Thank you**
> >
> > Thank you for your responses. I have no further questions.

---

### Official Review · Reviewer_cQSG · 2022-07-18

**Rating:** 7
**Confidence:** 2
**Soundness:** 3 good
**Presentation:** 3 good
**Contribution:** 3 good

**Summary:**

Information-Theoretic Generative Model Compression with Variational Energy-based Model focuses on compressing generative and structured prediction neural networks, using a combination of teacher student distillation (either online or offline) and a variational energy based model which is optimized by MCMC. Combining a flexible variational distribution rather than alternative approaches taken in prior work, this method shows improvement on several image to image and image generation network compression tasks when combined with existing methods in the literature.


**Questions:**

Are there specific technical issues preventing the application of this method to more recent (perhaps patch based, latent, or otherwise) models e.g. VQ-GAN, ViT-VQ-GAN, or other large scale image modeling networks used in recent text-to-image generation? If there are limitations in terms of assumptions, this would also be a good thing to state in a limitations-type section.

Similarly to the previous question, it seems like most (but not all) of the experiments compare with a best method + VID and/or CRD - particularly between Table 1 2 and 3, on the best performing methods sometimes CRD, VID, or both are missing. Is there a direct reason for this, beyond resources/time? "Filling the grid" so to speak, or explaining why VID/CRD is not applicable in some cases, would help follow-on researchers who may tackle one of these specific areas in greater detail.

**Limitations:**

The limitations (listed in section A.4) are fairly minimal, and finding other limitations of the method compared to baselines (listing increased hyperparameters / tuning / MCMC related issues for example) would be beneficial. Given application to a face-related model (StyleGAN2) there is potential to analyze or focus on a broader study of where and how the compression modifies generations. There are relevant studies (for example "Fairness for Image Generation with Uncertain Sensitive Attributes", Jalal et. al. or "Can Model Compression Improve NLP Fairness", Xu and Hu in NLP) which can provide direction for looking at subsets of the overall data, to assess FID or other structural metrics to determine if one subset is disproportionately changed by the compression compared to the baseline. If this study is overly involved (to the level of needing its own study/paper), it is also appropriate to state as a limitation that the compression method has not been extensively tested with respect to increasing, reducing, or matching the bias of the original model, only analyzing characteristics such as parameter count and aggregate performance.

**Strengths And Weaknesses:**

Strengths:
There are a wide swath of generative models and compression methods applicable to such networks, the authors have done a commendable job in ablations against other relevant methods, as well as detailing both the background / core methodology of their method and several others. Though the proposed method (in terms of implementation) is quite complex, the description as well as coupled code and detailed supplemental materials are sufficient to fully understand the proposed method. This clarity should make the paper accessible to the broader community.

Overall, I find the paper to be good quality, with significant interest to the model compression community and (with potential application to alternate methods) interest to the generative community at large.

Weaknesses:
Weaknesses in terms of experiments largely come in the form of "directional improvement", specifically for large, high performance models. The experiments here show a marked improvement in the small-to-medium scale, but application to more recent models could improve the impact of this work in the broader community - the lack of inference overhead, a large reduction in parameter count, and preservation or improvement of performance (in terms of FID) on the benchmarks are all critical aspects to running large scale text-to-image generators on commodity hardware.

One minor weakness is originality, as the proposed method follows closely in the footsteps of prior work from a methodology perspective, and sees gains in combination with other existing methods (rather than via replacement). This not a critical problem, but on the specific criteria of originality the contribution here feels "moderate", compared to other areas discussed in "Strengths".

A secondary weakness in the experiments is scale - more experiments on the scale of StyleGAN2 (or similar size models / image domains) would be potentially be beneficial. There are significant performance gains in terms of FID on the smaller scale experiments, but all methods (including this one) result in lower FID performance on the largest task.

While the proposed method does lower the gap between compressed and uncompressed (while having a drastic reduction in parameter count) compared to relevant benchmarks, it is still counter to the trend seen in several of the other experiments overall.

Having experiments on other domains or models (bedrooms as a potential intermediary? Large scale non-face models?) could show whether this effect is particular to StyleGAN2 (being a quite high performance model), or a broader trend on small scale vs large scale image models. Varied effects on small scale (potential performance *improvement* where parameter reduction is unneeded) and large scale (massive parameter reduction while minimizing the performance gap) experiments is a useful contribution, that is currently not spelled out. Though asking for more experiments is difficult given the extensive studies the authors have already provided, even one more large-scale experiment could potentially illuminate some properties of the proposed method. This bridges into some of the questions in the following section.

---

> ### Author Response · Authors · 2022-08-02
> **Author Response**
>
> We truly thank you for your constructive and positive comments and below are our responses to the main questions.
>
> Q1. Moderate originality
>
> A. We partially agree with your opinion since we follow the footsteps of prior works for fair comparisons and show the effectiveness of the proposed method when combined with them. However, we would like to emphasize that our key contributions lie in 1) the information-theoretic problem formulation of GAN compression using mutual information and 2) the introduction of EBMs to variational distributions and its successful application to a practical problem. Such ideas have not been addressed before and are technically sound, so we believe that our paper has sufficient novelty.
>
> Q2. Large-scale experiments
>
> A. We really appreciate you for a good suggestion, however, please understand that it is very difficult to perform the suggested large-scale experiments during the short rebuttal period.
>
> Q3. Are there specific technical issues preventing the application of this method to more recent models or other large scale image modeling networks used in recent text-to-image generation?
>
> A. Since the outputs are also images in text-to-image generation tasks, our method is easily applicable to the networks without any modifications.
>
> Q4. Missing experiments for CRD and VID in Table 1, 2, and 3
>
> A. Since we believe that it is enough to show that the proposed method outperforms CRD and VID as presented in Table 4,  we did not perform experiments for CRD and VID on all datasets and models. As suggested by Reviewer cQSG, we perform additional experiments except the Pix2Pix model using CRD due to the short rebuttal period while the results of the styleGAN2 experiments will be updated. Note that the results of Table A1 for the Horse $\rightarrow$ Zebra dataset are copied from Table 4 of the main paper.  As presented in table A1 and A2,  VEM is consistently more effective than the previous method while VID and CRD sometimes perform worse than the baseline. We appreciate you for a good suggestion and will add the experiments in the final version if our paper is accepted.
>
>
> Table A1: Performance comparison with CRD and VID using CycleGAN (related to Table 2 of the main paper).
>
> | Dataset | Method  | MACs | #Parameters  | FID ($\downarrow$) |
> |:----------------:|:---------------:|:---------------:|:---------------:|:---------------:|
> |    Horse $\rightarrow$ Zebra  |  OMGD*   | 1.41G (40.3x) | 0.14M (82.5x) | 57.14 |
> |   Horse $\rightarrow$ Zebra    |   OMGD* + VID   | 1.41G (40.3x) | 0.14M (82.5x) | 65.73 |
> |     Horse $\rightarrow$ Zebra     | OMGD* + CRD | 1.41G (40.3x) | 0.14M (82.5x) | 70.59 |
> |      Horse $\rightarrow$ Zebra      |  OMGD* + VEM   | 1.41G (40.3x) | 0.14M (82.5x) | 50.83 |
> |    Summer $\rightarrow$ Winter  |  OMGD*   | 1.41G (40.3x) | 0.14M (82.5x) | 75.20 |
> |   Summer $\rightarrow$ Winter    |   OMGD* + VID   | 1.41G (40.3x) | 0.14M (82.5x) | 74.39 |
> |      Summer $\rightarrow$ Winter     | OMGD* + CRD | 1.41G (40.3x) | 0.14M (82.5x) | 74.54 |
> |      Summer $\rightarrow$ Winter      |  OMGD* + VEM   | 1.41G (40.3x) | 0.14M (82.5x) | 74.04 |
>
> Table A2: Performance comparison with CRD and VID on the CelebA and FFHQ datasets using SAGAN and StyleGAN2, respectively (related to Table 3 of the main paper).
>
> |   Dataset   | Model | Method  | MACs | Compression Ratio  | FID ($\downarrow$) |
> |:----------------:|:---------------:|:---------------:|:---------------:|:---------------:|:---------------:|
> |    CelebA  | SAGAN |    GCC*   | 15.45M | 34.12%| 27.91 |
> |   CelebA    | SAGAN |  GCC* + VID   | 15.45M | 34.12% | 26.57 |
> |     CelebA   | SAGAN | GCC* + CRD | 15.45M | 34.12% | 30.93 |
> |      CelebA     |  SAGAN | GCC* + VEM   | 15.45M | 34.12%  | 25.27 |
> |    FFHQ  | StyleGAN2 |    CAGC*   | 4.10B | 90.91%| 7.90 |
> |   FFHQ    | StyleGAN2 |  GAGC* + VID   | 4.10B | 90.91% | 7.51 |
> |     FFHQ   | StyleGAN2 | CAGC* + CRD | 4.10B | 90.91% | 7.65 |
> |      FFHQ     |  StyleGAN2 | CAGC* + VEM   | 4.10B | 90.91%  | 7.48 |
>
> Q5. Discussion about limitations of the proposed method
>
> A. As you pointed out, our method increases the number of hyperparameters related to MCMC. Also, We thank you for your comment about the lack of an analysis about the bias problem for GAN compression approaches and the analysis seems to be a meaningful direction for the next research. We will add the limitations in the final version if our paper is accepted.

---

> > ### Author Response · Authors · 2022-08-06
> > **StyleGAN2 Experiments on the FFHQ dataset using CRD and VID**
> >
> > Dear Reviewer cQSG,
> >
> > The styleGAN2 experiments using VID and CRD on the FFHQ dataset are just finished and we present the results as below, where we also updated the missing results in Table A2.
> >
> > As a result, our method also outperforms the baseline methods.
> >
> > Thank you very much and please let us know if you have any questions.
> >
> >
> >
> > Table A3: Performance comparison with CRD and VID on the FFHQ dataset using StyleGAN2 (related to Table 3 of the main paper).
> >
> > |   Dataset   | Model | Method  | MACs | Compression Ratio  | FID ($\downarrow$) |
> > |:----------------:|:---------------:|:---------------:|:---------------:|:---------------:|:---------------:|
> > |    FFHQ  | StyleGAN2 |    CAGC*   | 4.10B | 90.91%| 7.90 |
> > |   FFHQ    | StyleGAN2 |  GAGC* + VID   | 4.10B | 90.91% | 7.51 |
> > |     FFHQ   | StyleGAN2 | CAGC* + CRD | 4.10B | 90.91% | 7.65 |
> > |      FFHQ     |  StyleGAN2 | CAGC* + VEM   | 4.10B | 90.91%  | 7.48 |

---

> > > ### Comment · Reviewer_cQSG · 2022-08-06
> > > **Reply to authors**
> > >
> > > These additional experiments enhance and strengthen the core arguments of the paper. Kudos to the authors for completing them in such a tight timeframe.
> > >
> > > Given the applicability of the method, as well as the array of experimental results I have updated my score on the paper. Thanks to the authors for the extra experiments, and answers to questions to myself and the other reviewers.

---

> > > > ### Author Response · Authors · 2022-08-08
> > > > **Author Response**
> > > >
> > > > Dear Reviewer cQSG,
> > > >
> > > > We appreciate you and will revise the main paper as discussed.
> > > >
> > > > Best wishes,
> > > >
> > > > Authors

---

### Author Response · Authors · 2022-08-10
**Summary of Revision**

We sincerely thank all reviewers for their constructive and positive comments and we present the summary of our responses to each reviewer as below.

Q1. Visual quality compared with the previous methods

A. Upon request by Reviewer tn1T, we present more qualitative results of VEM, OMGD, and CAGC in Figure 13, 14, 15 in the new supplementary document (Rebuttal_VEM.pdf). As seen in the presented images, VEM achieves qualitative better results than OMGD and CAGC and tends to generate the images that are visually similar to teacher images, which is also supported by Table A3. Note that we did not run additional experiments to prepare the new supplementary file and chose image pairs (out of the existing results made for our initial submission) that have larger visual quality differences.


Q2. Moderate Originality

A. We partially agree with your opinion since we follow the footsteps of prior works for fair comparisons and show the effectiveness of the proposed method when combined with them. However, we would like to emphasize that our key contributions lie in 1) the information-theoretic problem formulation of GAN compression using mutual information and 2) the introduction of EBMs to variational distributions and its successful application to a practical problem. Such ideas have not been addressed before and are technically sound, so we believe that our paper has sufficient novelty.


Q3. Additional comparisons with CRD and VID

A. As presented in Table A1, A2, VEM is consistently more effective than VID and CRD and the two methods sometimes perform worse than the baselines, which is also supported by Table 4 of the main paper.


Q4. Additional experiments for 1024x1024 resolutions using StyleGAN2

A. We appreciate you for a good suggestion, but it is really hard to perform the experiment you suggested since it takes an even longer time to compress a Stylegan2 for 1024 pixel image generation tasks. We believe that VEM can work well for generative models for 1024 pixels since our method is theoretically validated and not limited to the ones for 256 pixels. We will perform the experiment and will add the results in the final version if our paper is accepted.



Q5. The size of energy-based model and computational cost of VEM

A.  For the size of the energy-based model, it consumes 0.12G MACs for the Horse $\rightarrow$ Zebra dataset while the model requires 1.95G MACs for other datasets. Also, our method requires an extra 0.5x ∼1.5x training cost depending on datasets and models. Although the presence of MCMC in VEM incurs an extra training cost, the cost can be reduced by decreasing the number of MCMC steps. When the number of MCMC steps is reduced from 10 (used in all experiments) to 5 combined with OMGD, the FID score on the Horse $\rightarrow$ Zebra dataset is increased from 50.83 to 52.04, which still outperforms the baseline algorithm. Therefore, there is a trade-off between the performance and the training time and we will add the discussions in the final version.

Q6. Is VEM applicable in absence of a distillation method?

A. Yes, VEM is applicable when the previous methods are absent. But, we believe that VEM is more effective when combined with the methods.


Q7. Are there specific technical issues preventing the application of this method to more recent models or other large scale image modeling networks used in recent text-to-image generation?

A. Since the outputs are also images in text-to-image generation tasks, our method is easily applicable to the networks without any modifications.

Q8. Presentation issues related to clarification, Figure 1, limitations, and definition of abbreviations

A. We will carefully revise the paper to reflect all comments in the final version.

---

### Meta-Review · Area_Chair_XmBX · 2022-08-31

**Recommendation:** Accept
**Confidence:** Less certain

**Metareview:**

This work concerns the compression of generative adversarial networks and other image generation networks, such as dense prediction/image to image networks. Where existing approaches to compress these models rely on matching pairs of outputs, this work optimizes the Barber-Agakov lower bound on the differential mutual information between teacher and student, parameterizing the bound using an energy-based model. Offline distillation alternately fixes the student network parameters and the EBM parameters while optimizing the other, while online distillation can be performed by also including the teacher parameters, holding two of the three fixed while optimizing the third in a "round robin" fashion. The gradient of the EBM partition function is estimated using Langevin dynamics for a fixed number of steps, with chains initialized at student outputs. Qualitative and quantitative results support the assertion that this represents an improvement over existing approaches.

Reviewers found the paper overall quite clear, the method moderately original, the technical details sound, and the work well-situated in the context of other compression methods. Reviewer cQSG in particular praised the "commendable job in ablations against other relevant methods".  6hcq had several concerns around the clarity of the manuscript, which were addressed in rebuttal and hopefully can be incorporated into future versions. Several reviewers had concerns about the scale of experiments; the authors responded in rebuttal with megapixel experiments using StyleGAN2. The AC concurs with cQSG who remarked that these results significantly strengthen the paper.

Reviewer tn1T remained skeptical of the presented qualitative results, even after rebuttal; however, the authors were belatedly able to provide qualitative results on 1024x1024 images in the supplementary material. tn1T did not comment on these results, so I cannot infer how they feel about them, but they appear somewhat convincing to the AC: the samples presented from the VEM-trained do not show exhibit artifacts to the same degree.  There is, of course, the issue of cherry-picking, and I would encourage the authors to provide as many of these comparisons as possible at reduced size, highlighting artifacts where they appear but not dropping baseline samples which do not exhibit them (if such samples arise), and highlighting any noticeable artifacts in the images produced by their method.

The AC feels that this work is technically solid, and noting the endorsement of two reviewers and that the qualitative comparison demanded by tn1T was carried out but not acknowledged by tn1T, the case in favour of acceptance outweighs that in favour of rejection.

**Award:**

No

---

### Decision · Program_Chairs · 2022-09-14

Accept